# ADAR1 Promotes Myogenic Proliferation and Differentiation of Goat Skeletal Muscle Satellite Cells

**DOI:** 10.3390/cells13191607

**Published:** 2024-09-25

**Authors:** Zihao Zhao, Miao Xiao, Xiaoli Xu, Meijun Song, Dinghui Dai, Siyuan Zhan, Jiaxue Cao, Jiazhong Guo, Tao Zhong, Linjie Wang, Li Li, Hongping Zhang

**Affiliations:** Farm Animal Genetic Resources Exploration Innovation Key Laboratory of Sichuan Province, College of Animal Science and Technology, Sichuan Agricultural University, Chengdu 611130, China; z15528133630@163.com (Z.Z.); xmiao0816@163.com (M.X.); xuxiaoli02@163.com (X.X.); 2023102002@stu.sicau.edu.cn (M.S.); 71317@sicau.edu.cn (D.D.); siyuanzhan@sicau.edu.cn (S.Z.); jiaxuecao@sicau.edu.cn (J.C.); jiazhong.guo@sicau.edu.cn (J.G.); zhongtao@sicau.edu.cn (T.Z.); wanglinjie@sicau.edu.cn (L.W.)

**Keywords:** Adenosine deaminase acting on RNA (ADAR1), MuSCs, myogenic proliferation and differentiation, mRNA-seq, goat

## Abstract

As one of the most important economic traits for domestic animal husbandry, skeletal muscle is regulated by an intricate molecular network. Adenosine deaminase acting on RNA (ADAR1) involves various physiological processes and diseases, such as innate immunity and the development of lung adenocarcinoma, breast cancer, gastric cancer, etc. However, its role in skeletal muscle growth requires further clarification. Here, we explored the functions of ADAR1 in the myogenic process of goat skeletal muscle satellite cells (MuSCs). The ADAR1 transcripts were noticeably enriched in goat visceral tissues compared to skeletal muscle. Additionally, its levels in slow oxidative muscles like the psoas major and minor muscles were higher than in the fast oxidative glycolytic and fast glycolytic muscles. Among the two common isoforms from ADAR1, p110 is more abundant than p150. Moreover, overexpressing ADAR1 enhanced the proliferation and myogenic differentiation of MuSCs. The mRNA-seq performed on MuSCs’ knockdown of ADAR1 obtained 146 differentially expressed genes (DEGs), 87 upregulated and 59 downregulated. These DEGs were concentrated in muscle development and process pathways, such as the MAPK and cAMP signaling pathways. Furthermore, many DEGs as the key nodes defined by protein–protein interaction networks (PPI), including STAT3, MYH3/8, TGFβ2, and ACTN4, were closely related to the myogenic process. Finally, RNA immunoprecipitation combined with qPCR (RIP-qPCR) showed that ADAR1 binds to *PAX7* and *MyoD* mRNA. This study indicates that ADAR1 promotes the myogenic development of goat MuSCs, which provides a useful scientific reference for further exploring the ADAR1-related regulatory networks underlying mammal skeletal muscle growth.

## 1. Introduction

Skeletal muscles comprise ~40% of adult body weight and are crucial for animal health, locomotion, and metabolism. Myogenesis generates the multinucleated contractile myofibers and consequently governs skeletal muscle regeneration and growth [1]. As muscle stem cells, MuSCs (skeletal muscle satellite cells) are dispensable and undergo three sequential phases—proliferation, myogenic differentiation, and cell fusion—to form myotubes during their myogenic process [2,3,4]. It is well-known that muscle growth is driven by multiple coding genes, including *PAX7* [5], *PCNA* [6], *MyoD* [7], *MyoG* [7], *MyHC* [8], *MEF2C* [9], and *IGF1R* [10].

Given its role as a post-transcriptional modifier of mRNA, ADAR1 (Adenosine deaminase acting on RNA 1) has been spotlighted recently. It generates two typical protein isoforms—p150 (a 150 kDa protein) and p110 (a 110 kDa protein lacking a Zα domain) [11,12]—with distinct subcellular localization [13] and function [14,15]. In general, ADAR1 complicates post-transcriptional regulation by altering RNA sequences, consequently causing protein recoding [16] and miRNAs re-targeting [17]. Furthermore, ADAR1 can act as a factor of the functional axis to regulate various physiological processes [18,19]. Intriguingly, levels of ADAR1 mainly increase during active muscle re-differentiation, suggesting its involvement in myogenesis [20]. However, the underlying mechanisms of ADAR1 regulating skeletal muscle development remain unclear.

Here, we profiled the ADAR1 expression in goat tissues and MuSCs. Specifically, p110 and p150, two typical ADAR1 protein isoforms, exhibited tissue-specific expression. ADAR1 promotes the myogenic process of goat MuSCs, including proliferation and differentiation. Additionally, mRNA-seq analysis screened 146 differentially expressed genes (DEGs) affected by ADAR1. These DEGs are enriched in pathways associated with muscle development. Moreover, transcripts of *PAX7* and *MyoD* were bonded on ADAR1, indicating a pivotal role of ADAR1 in regulating goat MuSCs myogenesis. This research enhances our understanding of ARAR1-related mammal skeletal muscle development.

## 2. Materials and Methods

### 2.1. Animals and Samples Collection

The experimental goats were from Nanjiang Yellow Goats farm (Nanjiang County, Sichuan, China). As the first meaty goat breed artificially developed in China, Nanjiang Yellow Goats are well-known for their meat traits and adaptation. All goats were reared under standardized feeding and management conditions.

The embryonic goats were selected at 90 d and 120 d (E90, E120). All fetuses were obtained humanely by cesarean section. For the postnatal goats, three stages, 3 d, 60 d, and 120 d (B3, B60, and B120), were chosen. At each stage, three randomly selected female goats were humanely sacrificed and dissected. Samples from 6 muscles (longissimus dorsi muscle (LD), psoas major muscle, psoas minor muscle, supraspinatus muscle, gastrocnemius muscle, and quadriceps muscle) and five internal organs (including heart, spleen, lung, liver, and kidney) were collected, quick-frozen in liquid nitrogen, and then kept at −80 °C.

### 2.2. Cell Cultivating

MuSCs used were isolated from the LD of Nanjiang Yellow goats (3 d, female) in our laboratory [21]. We planted the MuSCs in the Growth medium (GM, high-glucose DMEM containing 10% FBS (Gibco, Grand Island, NY, USA) and 2% Penicillin and Streptomycin (Invitrogen, Carlsbad, CA, USA)) at 37 °C with 5 % CO_2_. Once their confluence reached 80% to 90%, they were digested in 0.25 % trypsin and, after that, mixed and cultivated in 6-well plates (1 × 10^4^~10^5^ per well). To induce myogenic differentiation, a differentiation medium (DM, harboring 2% HS (Gibco) and 2% Penicillin and Streptomycin) replaced the GM when cells amounted to 80%–90% confluence.

### 2.3. RNA Extraction from Cells and Tissues

We extracted total RNAs from cells or tissues utilizing the RNAiso Plus (TaKaRa, Kusatsu, Japan) and then purified it using RNeasy Mini Kit (QIAGEN, Chatsworth, CA, USA). To assess their integrity and any contamination, total RNAs were determined by electrophoresis on 1.5% agarose gel. Subsequently, the RNAs’ purity and concentration were further quantified using a NanoPhotometer^®^ (IMPLEN, Los Angeles, CA, USA) and Qubit RNA Assay Kit in a Qubit^®^ 2.0 Fluorometer (Life Technologies, Carlsbad, CA, USA), respectively. Those qualified RNAs were then securely stored at −80 °C.

### 2.4. Construction of ADAR1 Overexpression Plasmid

In this experiment, the pEGFP-N1 (Promega, Madison, WI, USA) was selected to construct an ADAR1 overexpression plasmid, and DH5α competent cells (Tsingke, Beijing, China) were employed for the transformation of the recombinant plasmid. After amplifying the full-length fragment of ADAR1 CDS (XM_018046195, including isoforms p150 and p110), restriction enzyme cleavage sites, Kpn 1 (GGTACC) and BamH1 (GGATCC), and protective bases (CGG) were added to both ends of the fragment. Following enzymatic digestion, the target sequence was ligated with the linearized empty vector using T4 DNA ligase (Takara, Dalian, China). The ligation product was then transformed into competent cells and inoculated into the liquid LD medium for bacterial strain preparation. Subsequently, 50 μL of the bacterial strain was spread onto the solid LD medium containing kanamycin, and after inverted incubation for 12 h, a single clone was picked and propagated in the LD liquid medium. We extracted the recombinant plasmid using the EasyPure HiPure Plasmid MiniPrep Kit (TransGen, Beijing, China). Sanger sequencing (Tsingke, Beijing, China) was performed on the plasmid to ensure complete sequence accuracy.

### 2.5. Library Construction, mRNA Sequencing and Data Processing

The RNA-seq for this experiment was entrusted to Beijing Novogene Co., Ltd. (Beijing, China). The Fragment Analyzer 5400 (Agilent Technologies, Santa Clara, CA, USA) was used to evaluate the RNA integrity. Then, the qualified RNA (1 μg for each sample) was consumed for library construction. To guarantee the quality of the library constructed, an initial quantification was carried out using the Qubit 2.0 system, followed by detecting the size of the insert fragment on the Agilent 2100 platform; then, the effec tive concentration of the library was precisely measured using qPCR. Finally, the qualified library was sequenced on Illumina NovaSeq 6000 platform.

The clean reads were yielded by filtering through the raw data, rigorously excluding those containing adapters, >10 % poly(N) content, or over half bases scoring < a Phred threshold 10. Using the goat reference genome and associated gene model annotation files retrieved from NCBI database (CHIR_1.0), we built a reference genome index with Bowtie v2.0.6, and aligned the paired-end clean reads to the reference genome by TopHat v2.0.14. Moreover, utilizing RABT (the reference annotation-based transcript) in Cufflinks version 2.2.1, we constructed and identified mRNA transcripts from the alignment outcomes produced by TopHat2. Subsequently, a comprehensive view of the expression characteristics was conducted via hierarchical clustering, based on the differential expression values of the transcripts. We used the DESeq2 R package (version 1.42.1) for differential read counts analysis to identify DEGs (|log2 Fold Change| > 0.1, and padj < 0.05). GO and KEGG were analyzed using ClusterProfile R package (version 4.10.1) (padj < 0.05).

### 2.6. RT-qPCR Analysis

We used the HiScript Q RT SuperMix (Vazyme, Nanjing, China) reverse RNA (1 μg RNA) to cDNA. ChamQ SYBR Color qPCR Master Mix (Vazyme, China) was adopted in RT-qPCR with the following reaction system: 5 μL SYBR, 3.4 μL ddH_2_O, 0.4 μL forward/reverse primer (10 μM), and 0.8 μL cDNA. The PCR programs performed on the Bio-Rad CFX96 (Bio-Rad, USA) included denaturation for 30 s at 95 °C, followed by 40 cycles for 20 s at 95 °C, for 20 s at 60 °C, and 30 s at 72 °C. To determine the relative mRNA levels of a target gene, we employed the 2^−ΔΔCt^ method with GAPDH as an internal control. The primers employed for performing RT-qPCR are detailed in Appendix A.

### 2.7. Western Blot (WB) and Antibodies

Using the total protein extraction kit (Bestbio, Nanjing, China), we extracted the total protein from tissues and cells and stored it at −80 °C. The 5 × loading buffer was mixed with the total protein and then denatured at 95 °C for 5 min.

WB used the following primary antibodies: ADAR1 (1:200, Santa CruZ, Dallas, TX, USA, sc-271854), *PAX7* (1:200, Santa CruZ, sc-81648), *PCNA* (1:200, Santa CruZ, sc-56), *MyoD* (1:1000, Abcam, Cambridge, MA, USA, ab133627), *MyoG* (1:1000, Abcam, ab1835), *MyHC* (1:200, Santa CruZ, sc-376157) and β-tubulin (1:5000, ZENBIO, Chengdu, China, 250007). The secondary antibodies are as follows: anti-mouse IgG (1:5000, ZENBIO, 511103) and anti-rabbit IgG (1:5000, ABclonal, Wuhan, China, AS014). Image J (version 1.54) was used for semi-quantitative analysis, and the target proteins were normalized to β-tubulin.

### 2.8. EdU Assay for Proliferating Cells

According to the instructions of the BeyoClick™ EdU-488 kit (Beyotime, Shanghai, China), we performed steps such as the fixation, permeabilization, and staining of proliferating cells. At least three images for each group were captured by a fluorescence microscope (Olympus, Tokyo, Japan). The EdU^+^ cell % equals the percentage of the number of EdU-positive nuclei divided by the total number of nuclei presented in the image. The nuclei analyzed were over 1.5 × 10^4^.

### 2.9. CCK-8. Analysis of Cell Number

We planted the MuSCs (1 × 10^4^~10^5^ cells per well) into a 96-well plate. When processed at 60% to 70% confluence, cells were configurated 10% CCK-8 incubation solution (GM: CCK-8 = 9:1), replaced GM with the 10% CCK-8 solution and incubated for 2 h. We detected the 450 nm absorbance by MultiskanGO (Thermo Fisher, Waltham, MA, USA).

### 2.10. Immunofluorescence Analysis for MyHC

After removing the DM, we rinsed the differentiating cells with ice-cold PBS, then fixed them with 4% paraformaldehyde for 15 min. Furthermore, we permeabilized them with 0.5% Triton X-100 at 4 °C for 10 min, blocked with 2% BSA at 37 °C for 30 min. Then, cells to which the *MyHC* antibody was added (1:200, Santa CruZ) were incubated overnight at 4 °C, followed by incubating with a secondary antibody, Cy3_IgG (1:200, Solarbio, Shanghai, China), at 37 °C for 2 h. Ultimately, the cells were stained with DAPI (0.05 g/mL) at 37 °C for 10 min in a light-prohibited container. We examined the images under a fluorescent microscope (Olympus, Tokyo, Japan). ImageJ (version 1.54) was meticulously utilized to count cells, determining the nuclei (DAPI channel) and nuclei surrounded by the *MyHC*. Subsequently, the percentage of *MyHC*^+^ was computed by dividing the nuclei number encircled by the *MyHC* signal by the total count of nuclei. This process was performed independently for at least three repeats per treatment, with five randomly chosen areas analyzed for each sample.

### 2.11. RNA Immunoprecipitation (RIP) Assay

The Magna RIP RNA-Binding Protein Immunoprecipitation Kit (univ, Shanghai, China) was used for RIP assays with antibodies ADAR1 (1:20, Santa Cruz) and IgG. According to the manufacture’s guideline, we washed the proliferation or differentiation cells (~1.5 × 10^7^) with PBS, then digested them with trypsin, resuspended, and centrifuged them to isolate the cell’s precipitate. Then, the cells were lysed using 250 μL buffer, and incubated with antibodies overnight, and the remaining steps were performed accordingly.

### 2.12. Statistical Analysis

The data are presented as the mean ± SEM and all statistical analyses were conducted utilizing GraphPad Prism software (version 6.01, GraphPad Software, San Diego, CA, USA). Unpaired Student’s *t*-tests were employed to assess the differences in the means between the two groups. To analyze the mean differences among three groups, ANOVA was utilized, with a Tukey test conducted for multiple comparisons. Statistically significant levels were set as * *p* < 0.05, ** *p* < 0.01, and ns *p* > 0.05.

## 3. Results

### 3.1. Homology and Subtypes of Goat ADAR1 Protein

To gain a deeper understanding of ADAR1, we first examined its protein homology. Using MEGA 11 software, we constructed a phylogenetic tree, showing that the goat (*Capra hircus*) ADAR1 protein closely homologized to sheep (*Ovis aries*) and distantly to humans (*Homo sapiens*) and mice (*Mus musculus*) (Figure 1A). In addition, we found that the homology of pigs’ (*Sus scrofa*) ADAR1 protein is relatively distant from other even-toed ungulates such as sheep, goats, antelopes (*Oryx dammah*), and cattle (*Bos taurus*), which may be due to the lack of a low complexity region after the third double-stranded RNA-binding motifs (DSRM), although these functional structures are highly conserved across species (Figure 1B).

Moreover, seven subtypes potentially originated from goat ADAR1. Among them, subtypes X1 to X4 have a high degree of amino acid similarity, and subtypes X5–X7 are identical (Appendix A). So, we used subtypes X1 (comprised of 1175 amino acids and similar to p150 in other species) and X5 (similar to p110) for further analysis. We used the online tool SMART to predict the secondary structure of ADAR1 protein. Both p150 and p110 harbor three double-stranded RNA-binding motifs (DSRM) and one ADEAMc domain. In particular, p150 possesses two Z-DNA-binding domains, one more than p110 (Figure 1C). Additionally, the tertiary structure of p150 and p110 were distinguished (Figure 1D).

### 3.2. Expression Profiles of ADAR1 in Goat’s Tissues and Cells

We profiled ADAR1 mRNA and protein in goat’s tissues. The results showed that ADAR1 mRNA in goat muscles was significantly lower than in visceral tissues (*p* < 0.05, Figure 2A). The total ADAR1 protein was highly enriched in the liver, followed by muscle, and lowest in the kidney (Figure 2B). It is well-known that p110 and p150 are the main protein subtypes encoded by ADAR1 [22]. We found that p110 was highest in goat liver and lowest in the kidney, and p150 was enriched in non-muscle tissues detected, such as the liver and spleen (Figure 2B). These results suggest that ADAR1 abundance is tissue-specific in goats, and p110 is more prominent in muscles than p150.

Furthermore, we detected the ADAR1 in six different muscles. Generally, ADAR1 mRNA was most enriched in slow oxidation muscle (the psoas major and psoas minor muscles), followed by fast glycolytic (the longissimus dorsi and supraspinatus muscles) and fast oxidative glycolytic muscle (the gastrocnemius and quadriceps muscles). With muscle development, ADAR1 transcripts in fast muscles were slowly increased and were highly enriched in the longissimus dorsi at B120 (Figure 2C). Similarly, ADAR1 protein was highest in the psoas major and relatively high in the longissimus dorsi muscle (Figure 2D). Intriguingly, p150 was almost consistent in differed muscle, with p110 altered according to muscle type. Moreover, p110 was generally more abundant than p150 within the same tissue (Figure 2D).

We further profiled the mRNA and protein of ADAR1 during the myogenesis of goat MuSCs. The mRNA of ADAR1 continued to increase from proliferation to myogenic differentiation and peaked at 168 h (Figure 2E), similar to the ADAR1 protein. ADAR1 was higher in differentiating cells than in the proliferation stage and peaked at 72 h of differentiation (Figure 2F). Notably, p110 was enriched in differentiating cells, while p150 was almost undetectable (Figure 2F).

In summary, ADAR1 is ubiquitously expressed in goat tissues, and intrinsic p110 is likely more functional than p150 in goat myogenesis.

### 3.3. ADAR1 Promotes Proliferation and Differentiation of Goat MuSCs

To explore the role of ADAR1 in goat MuSCs, we constructed plasmid overexpressing ADAR1 (p-ADAR1) and designed siRNA targeting ADAR1 (si-ARAD1), and found 5 μg and 100 nM are the optimal concentrations for experiments, respectively (Appendix A). The ectopic ADAR1 resulted in a notable enhancement in its levels, encompassing both the catalytic and RNA-binding domains, confirming the successful overexpression of the full-length ADAR1 CDS (Appendix A). Although the levels of *PAX7* and *PCNA* transcripts did not change significantly after ADAR1 overexpression, their protein levels, especially *PCNA*, were significantly elevated. Expectedly, inhibiting ADAR1 dramatically decreased *PAX7* mRNA (Figure 3A) and both *PAX7* and *PCNA* proteins (Figure 3B), though *PCNA* mRNA was insignificantly changed. ADAR1 overexpression significantly promoted absorbance at 450 nm after 24 h, 48 h, and 72 h assayed by CCK-8 (Figure 3C) and the generation of newly formed cells evaluated by EdU (Figure 3D). These findings suggest that ADAR1 enhance the proliferation of goat MuSCs.

We further disturbed ADAR1 in differentiating cells. After inhibiting ADAR1, the levels of *MyoG*, *MyHC*, and *Myomaker* mRNA were significantly decreased (Figure 3E). Consistently, the protein of *MyoG* (Figure 3F) and *MyHC* (Figure 3G) were downregulated. Meanwhile, an overexpression of ADAR1 failed to change the mRNA of differentiation marker genes (*p* > 0.05), but the *MyoG* proteins showed an increased trend (Figure 3E–G). *MyHC* immunofluorescence staining experiments showed that myotubes’ fusion rate and area decreased in cells with a decrease in ADAR1. The opposite result was observed when ADAR1 was overexpressed (Figure 3H,I). These findings imply that ADAR1 advanced the myogenic differentiation of goat MuSCs.

### 3.4. Genes Affected by ADAR1 in Differentiating MuSCs

To investigate the mechanism underlying ADAR1 in the myogenic differentiation of goat MuSCs, we conducted mRNA-seq analysis on cells inhibiting ADAR1. A total of 47.18 G of clean data were obtained. The proportion of bases in Q20 and Q30 was more than 97% and 92%, indicating that the sequencing quality was good (Appendix A). Additionally, the robust reproducibility of samples within identical groups in our experiments was shown by applying Hclust hierarchical clustering and PCA analysis (Appendix A). Moreover, we identified 146 DEGs, with 87 genes upregulated and 59 genes downregulated (Padj < 0.05, Log2FC > 0.1, Figure 4A). Specifically, the decrease in ADAR1 altered the expression of multiple myogenic factors, including *MYH3/8* [23,24], *IGFBP5/6* [25,26], *TGFβ2* [27], *CAPN6* [28], *TNNC1* [29], *TNNI1* [30], *MYOM1* [31] (Appendix A). Subsequently, we randomly selected five upregulated and five downregulated DEGs and quantified their levels using RT-qPCR. The outcomes obtained were in alignment with the findings from the RNA-seq analysis (Figure 4B,C). We employed the online tools STRING and Cytoscape and determined that STAT3, MYH3/8, TGFβ2, and ACTN4 were key nodes related to the myogenic process (Appendix A).

We further used GO and KEGG analyses to investigate DEGs’ functions. Many GO terms enriched for the downregulated genes were related to muscle development, including muscle contraction, muscle organ development, muscle system process, muscle tissue development, myofibers, myosin complex, and structural constituent of muscle (Figure 4D). Meanwhile, the upregulated genes were mainly enriched in ameboid-type cell migration, cell–substrate adhesion, and basement membrane (Figure 4E). The KEGG analysis showed that some classical muscle-related pathways, incorporating the MAPK signaling pathway, cardiac muscle contraction, and cAMP signaling pathway, were among the top 20 in the downregulated genes (Figure 4F). In contrast, those upregulated genes were mainly concentrated in pathways related to proteoglycans in cancer, gastric cancer, and steroid biosynthesis (Figure 4G). In summary, these results confirmed that ADAR1 positively affects various functions and pathways related to muscle development.

### 3.5. ADAR1 Binds on PAX7 and MyoD Transcripts

To further explore the mechanism of ADAR1 in MuSCs, we found that ADAR1 potentially binds to *PAX7* and *MyoD*, as predicted by the online software catRAPID (Figure 5A,B). To confirm it, we performed RIP-qPCR using RNA extracted from the proliferating and the differentiating cells. As expected, *PAX7* and *MyoD* mRNA were significantly enriched by the ADAR1 antibody, comparing with the control IgG (Figure 5C). We further employed gel electrophoresis to detect the amplicons of RIP-qPCR for *PAX7* and *MyoD*. The results indicated that ADAR1 successfully pulled down the mRNA of *PAX7* and *MyoD*, suggesting that the ADAR1 protein efficiently binds to transcripts of both *PAX7* and *MyoD* (Figure 5D,E), in line with the established fact that ADAR1 is an RNA-binding protein (RBP).

## 4. Discussion

ADAR1, a dsRNA-binding protein, is crucial in numerous biological processes, including tumor growth, immune response, embryonic development, and cell apoptosis [32]. Its transcripts and proteins have been identified in diverse species ranging from humans to model animals and even certain economic animals [33,34]. Our research has unveiled that the ADAR1 protein exhibits dsRNA binding and possesses catalytic domains across diverse species, implying a strong conservation of its functional properties.

In contrast to ADAR2 and ADAR3, ADAR1 is a ubiquitously expressed RNA-binding protein with tissue and species specificity. For instance, in fish, the mRNA levels of the ADAR1 (CiADAR1) gene are elevated in blood and spleen but reduced in muscle and kidney tissue [35]. Conversely, in mice, ADAR1 (ADAR) is predominantly expressed in the brain and spleen, while its levels are lower in the testes and skin [36]. In this study, we found higher levels of ADAR1 in the spleen and liver, with significantly lower muscle levels than other tissues.

Skeletal muscle is highly adaptable in response to different stimuli [37]. The slow oxidation, fast oxidative glycolytic, and fast glycolytic muscle all possess distinct characteristics [38]. In this study, ADAR1 expression in different muscle types varied little over time, except in the longissimus dorsi. Notably, slow oxidation muscle fibers expressed higher levels of ADAR1 in the mid-late embryogenesis (E90, E120) and neonatal periods (B3), surpassing fast glycolytic and fast oxidative glycolytic muscle types throughout growth. During the embryonic stage, muscle fiber numbers primarily proliferate and stabilize after birth [39]. However, in obesity, injury, or nerve compression, the three distinct types of muscle fibers can potentially undergo mutual transformation [40,41,42]. ADAR1 correlates with embryo muscle fiber development, maturation, and transformation, yet its precise mechanisms await further exploration.

ADAR1 produces two distinct subtypes, namely p150 and p110. Notably, the p150 subtype dynamically shuttles between the nucleoplasm and cytoplasm, with a predominant localization in the latter. In contrast, p110 primarily resides in the nucleus [43]. IFN exhibits heightened expression in the immune organs of mammals, and its abundance significantly increases in visceral tissues like the liver and lungs in response to stimuli. Conversely, its expression in muscle tissues remains relatively low under normal conditions [44]. Our research revealed that the level of the ADAR1 p150 subtype surpasses that of the p110 subtype in organs such as the liver, spleen, lungs, and kidneys. Conversely, in muscle tissues and MuSCs, the protein of p110 is consistently higher than that of p150, reinforcing the positive regulatory link between the p150 subtype and IFN. Furthermore, the expression patterns p150 and p110 mirror total ADAR1 across different tissues, demonstrating a consistent tissue specificity. Notably, the p150 transcript, which harbors the p110 promoter, can concurrently express both the p150 and p110 subtypes [11]. Our recombinant plasmid, derived from transcript X1 (XM_018046195.1), facilitates the simultaneous detection of these ADAR1 protein subtypes post-transfection.

ADAR1 functions as both an RNA-editing and an RNA-binding enzyme, exhibiting editing-dependent and editing-independent capabilities [45]. Specifically, ADAR1 has been well-documented to proliferate various cells, including tumor cells [33,46,47], preadipocytes [48], and C2C12 myoblasts [19]. Conversely, ADAR1 inhibits differentiation in porcine preadipocytes [48] while promoting differentiation in C2C12 myoblasts [19]. Intriguingly, our study reveals that ADAR1 expands both proliferation and differentiation in goat MuSCs. This disparity could be attributed to its species and/or tissue-specific mechanisms, and also suggests that ADAR1 plays an essential role in the development of goat MuSCs.

Upon interfering with ADAR1 in preadipocytes, researchers observed that the DEGs were predominantly enriched in key signaling pathways, including MAPK and cAMP [48]. Notably, these crucial pathways were also significantly enriched in our present research, indicating a consistent trend with previous findings. Specifically, mRNA-seq analysis following ADAR1 inhibition revealed significant alterations in multiple myogenesis factors, with the DEGs notably enriched in diverse myogenesis functions and pathways. Furthermore, the PPI network of these DEGs highlighted key nodes that encompassed crucial myogenic factors, underscoring the pivotal role of ADAR1 in the myogenic development of goats. The RIP-qPCR experiment validates that ADAR1 interacts with *PAX7* and *MyoD*, suggesting ADAR1 potentially influences the myogenic process of MuSCs by binding on transcripts.

## 5. Conclusions

We successfully profiled the ADAR1 expression pattern in tissues and muscles of goats. We found that ADAR1 promotes the myogenic process of goat MuSCs. Alterations in ADAR1 expression trigger variations in muscle development-related factors and pathways, highlighting its crucial role in muscle growth. Mechanistically, ADAR1 can bind with *PAX7* and *MyoD*, thereby modulating the development of MuSCs.

## Figures and Tables

**Figure 1 cells-13-01607-f001:**
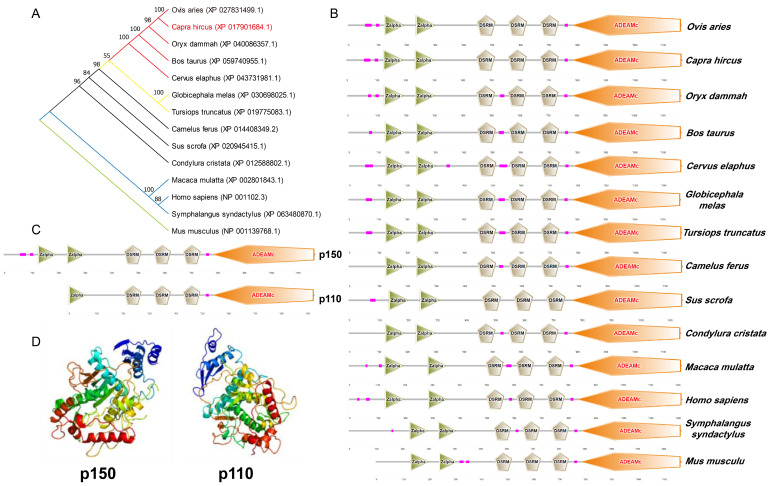
Homology and protein subtypes analysis of goat ADAR1. (**A**) Phylogenetic tree of ADAR1 among species. (**B**) ADAR1 protein domains among species. (**C**) Secondary structure of two isoforms (p150 and p110) of goat ADAR1. (**D**) Tertiary structure of goat p150 and p110.

**Figure 2 cells-13-01607-f002:**
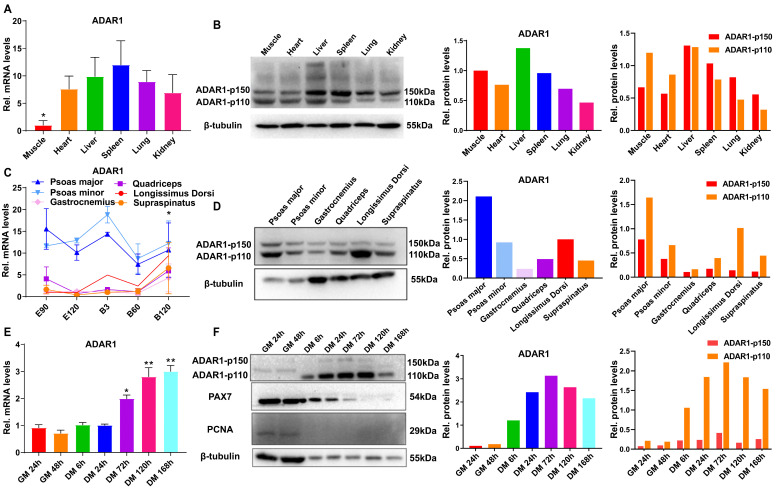
ADAR1 expression patterns in goat tissues and cells. (**A**) ADAR1 mRNA levels in various tissues of goat. (**B**) ADAR1 protein in various tissues of goat. (**C**) ADAR1 mRNA in six muscles. (**D**) ADAR1 protein in six muscles. (**E**) ADAR1 mRNA levels at the GM and DM stages of goat MuSCs. (**F**) ADAR1 protein at the GM and DM stages of goat MuSCs. Each qPCR experiment contained three biological replicates. * *p* < 0.05, ** *p* < 0.01.

**Figure 3 cells-13-01607-f003:**
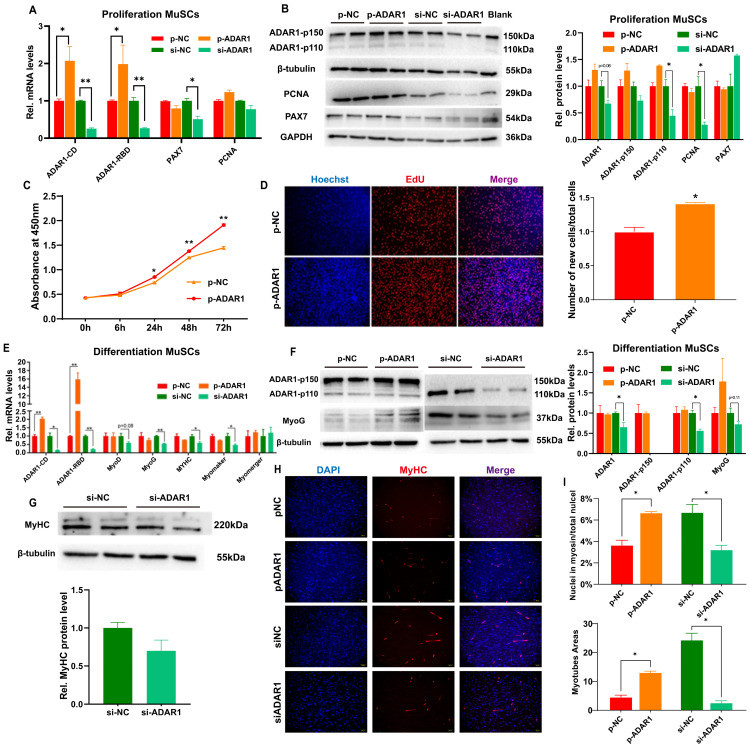
ADAR1 promotes the proliferation and differentiation of goat MuSCs. (**A**) The mRNA levels in proliferating cells overexpressing and inhibiting ADAR1. (**B**) The protein in proliferating cells overexpressed and inhibited ADAR1. (**C**) CCK-8 analysis of proliferating MuSCs overexpressed ADAR1. (**D**) EdU analyses after overexpressing ADAR1 in proliferating MuSCs. (**E**) The mRNA expression after overexpressing and inhibiting ADAR1 in differentiating cells. (**F**,**G**) The proteins in differentiating cells disturbed ADAR1. (**H**,**I**) Immunofluorescence analyses of myotubes after overexpressing and inhibiting ADAR1 in MuSCs’ differentiation stage. Each qPCR experiment contained three biological replicates. * *p* < 0.05, ** *p* < 0.01.

**Figure 4 cells-13-01607-f004:**
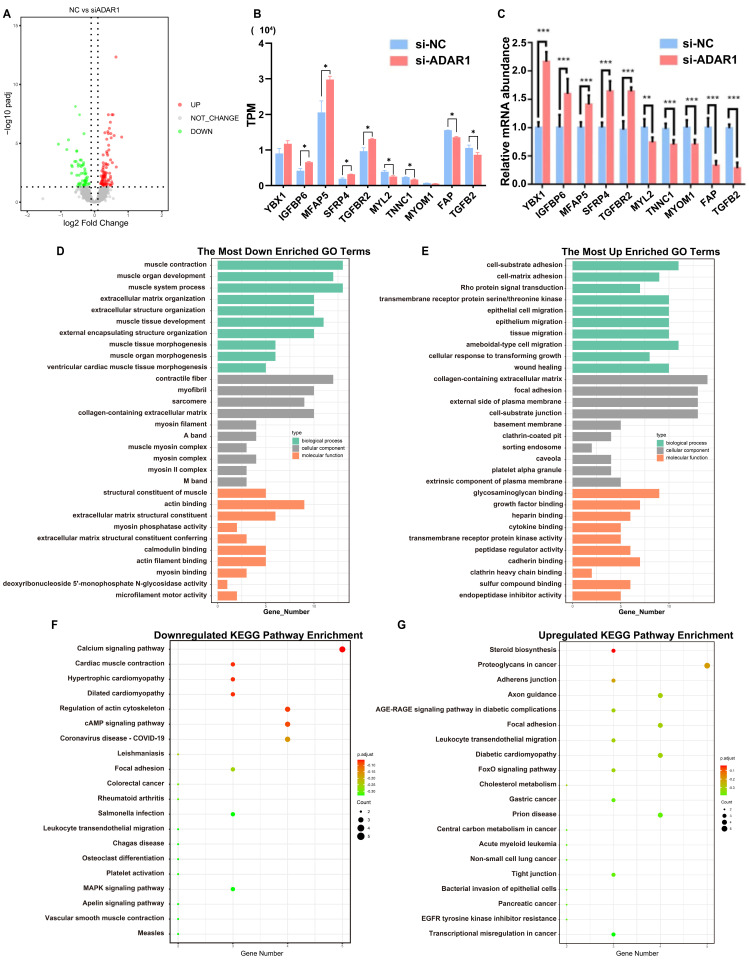
Genes affected by ADAR1 in myogenic differentiation of goat MuSCs. (**A**) Volcano map of DEGs. (**B**) TPM of randomly selected DEGs in mRNA-seq (5 up- and 5 downregulated). (**C**) mRNA levels of randomly selected DEGs quantified by RT-qPCR (5 up- and 5 downregulated). (**D**,**E**) GO enrichment analysis of the downregulated (**D**) and upregulated (**E**) DEGs. (**F**,**G**) KEGG signaling pathway analysis of downregulated (**F**) and upregulated (**G**) DEGs. Each qPCR experiments contained three biological replicates. * *p* < 0.05, ** *p* < 0.01, *** *p* < 0.001.

**Figure 5 cells-13-01607-f005:**
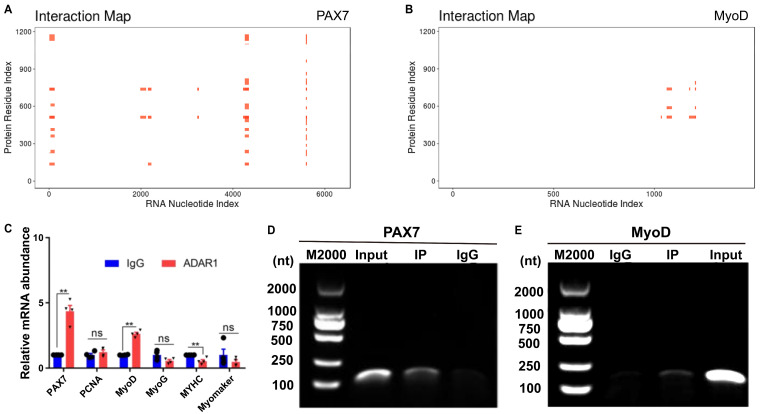
ADAR1 interacts with *PAX7* and *MyoD* mRNA in goat cells. (**A**,**B**) The binding capacity of ADAR1 protein on *PAX7* or *MyoD* transcripts was predicted using the online website catRAPID. (**C**) *PAX7*, *PCNA*, *MyoD*, *MyoG*, *MyHC*, and Myomaker mRNA enriched by anti-ADAR1 (IP, red) and anti-IgG (negative control, blue). (**D**) Gel electrophoresis of the *PAX7* mRNA detected for RIP-qPCR. (**E**) Gel electrophoresis of the *MyoD* transcripts detected for RIP-qPCR. Each qPCR experiments contained three biological replicates. ns *p* > 0.05, ** *p* < 0.01.

## Data Availability

The raw sequence data reported in this paper (GSA: CRA017927) have been deposited in the Genome Sequence Archive (Genomics, Proteomics & Bioinformatics 2021) at the National Genomics Data Center, China National Center for Bioinformation/Beijing Institute of Genomics, Chinese Academy of Sciences and publicly accessible at https://ngdc.cncb.ac.cn/gsa (accessed on 22 September 2024).

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
