# Peer review of "ADAR1 Promotes Myogenic Proliferation and Differentiation of Goat Skeletal Muscle Satellite Cells"

_cells, 2024, doi:10.3390/cells13191607_

Round 1
Reviewer 1 Report
Comments and Suggestions for Authors
Zhao et al. assessed the role of ADAR1 in goat skeletal muscle. To this end, they assess a number of muscles at different developmental stages and performed an in vitro study using goat muscle stem cells with overexpression / downregulation of ADAR1.
The study appears to be well executed, but the manuscript needs more details. The discussion needs more in depth and critical discussion of results. Please see below my specific comments.
Line 11 be more specific than “ADAR1 involves various physiological processes and diseases”
Line 22 RIP-qPCR, full name needed
Line 47 Elaborate more on the isoforms p110 and p150
Line 56 Specify which, Nanjing brown goats line 67?
Line 122-127 how much RNA was used for cDNA synthesis, how much input for each qPCR reaction, concentration primers in qPCR?
Line 143 add minimum number of nuclei per sample that were analysed
2.11 needs more detail regarding input amount, incubation times…
Line 239 mention fold increase (mean+/-sd)
Figures and Figures legends: mention if error bars are sd or sem, * p<0.05?, and number of biological replicates analysed. Some graphs can made a bit more clear by including a bit more information on axis. E.g. fig 3a and fig 3c, include proliferating MuSc and differentiating MuSc respectively on axis. Why are some figures in colors and others in black and white. Choose one color scheme and be consistent.
Line 245. Shortly explain what you assess with cck-8 assay and Edu.
Figure 3c. y-axis absorbance or relative change? Error bars? Add meaning of "**" to figure legend
3.3 / Figure 3f/g. Please include more details. At which time-point during differentiation were cells analysed? Given that proliferation was increased in p-ADAR1 cells, was the timepoint for switching to differentiation medium earlier or at the same time as p-NC? At differentiation analysis, where there more total nuclei in p-ADAR1 wells than in p-NC1 wells?
3.4. More information is needed: which timepoint analysed? How many days after transfection? How many replicates per group? Control group si-NC? What was ADAR1 expression in this experiment? Include in text what TPM (figure 4b) is. What is minimal fold change for DEG? What was maximum fold change (highest increase and strongest decreased gene in si-ADAR1?
3.5 / figure 5c. which genes were analysed in proliferating MuSC and which one in differentiating MuSC and at which timepoint?
Line 328 specify if you mean RNA or protein expression. In addition, elaborate on your observation that ADAR1 RNA expression in muscle was ~10? Fold lower than e.g. spleen, but protein expression was the same between muscle and spleen.
Line 333-335 Interesting finding, discuss why this may be the case.
Line 341-348 these sentences should be moved to introduction
Line 352 elaborate on the link to IFN and muscle
Line 358-369 Figure 3 -5 and accompanying sections in results contains huge amount of data, yet only ~10 lines in discussion section, which don’t even contain in depth discussion, but merely a summary of the main findings. This section needs more discussion, comparison to other data, strengths and weaknesses of the study. Amongst others, authors need to discuss why % of myosin+ nuclei is the same in pADAR1 and si-NC and siADAR1 is similar to pNC (figure 3i) and myotube area in siNC is the highest of all four groups. Authors conclude that ADAR1 stimulates both muscle stem cell proliferation and differentiation. Discuss how this would mechanistically work, as many genes in literature have been shown to either stimulate proliferation or differentiation, but not both. Discuss if the increased proliferation affected study procedures (see also above comment on details in results section). Discuss other papers on role of ADAR1 in myogenesis, eg. Hsieh et al. Cell Death and Differentiation 2014, that studied ADAR1 knockdown in C2C12 myogenic stem cells.
Comments on the Quality of English LanguageIn general fine, some rephrasing by a native English speaker would improve clarity.
Reviewer 2 Report
Comments and Suggestions for Authors
The study by Zhao et al. reports on the role of ADAR1 in the myogenic differentiation of skeletal muscle stem cells. They performed a comprehensive set of experiments and presented the results that strongly validate their conclusions. The subject of the report is highly relevant to the field. However, the major weakness of the manuscript is the written text which needs extensive revision.
1) The manuscript is extremely poorly written. There are numerous confusing sentences, phrases, and words throughout the manuscript which make it difficult to understand the study content or follow the conclusions. Because there are too many, I marked the sections which needs rigorous revision in the attached PDF. I strongly recommend more careful editing of the manuscript text and encourage the authors to use professional editing services.
2) The figures are poorly explained. The reader needs to guess what the labels really mean.
Fig. 2A, C, and E, please indicate which RNA (y axis).
Fig. 3A and E, please indicate which RNA (y axis).
Fig. 3, please explain p-NC, p-ADAR1, si-NC, si-ADAR1, and Blank
Fig. 4C, please indicate which RNA (y axis).
Fig. 5C, please indicate which RNA (y axis).
4) Research involving animals or animal samples must have been performed in accordance with the Declaration of Helsinki and approved by a local ethics committee. Please add “The study was conducted according to the guidelines of the Declaration of Helsinki and approved by the Institutional Review Board xxx of xxx.”

I'd like to encourage the authors to use professional editing services.
Round 2
Reviewer 2 Report
Comments and Suggestions for Authors
Please introduce the abbreviation WB in line 144 as follows:
2.7 Western blot (WB) and antibodies
Author Response
Comment 1: Please introduce the abbreviation WB in line 144 as follows:
2.7 Western blot (WB) and antibodies
Response 1: Thank you for your suggestion. We have made modifications to the revised manuscript accordingly. (Line 139)